# Lipid Aldehydes 4-Hydroxynonenal and 4-Hydroxyhexenal Exposure Differentially Impact Lipogenic Pathways in Human Placenta

**DOI:** 10.3390/biology12040527

**Published:** 2023-03-30

**Authors:** Aisha Rasool, Taysir Mahmoud, Perrie O’Tierney-Ginn

**Affiliations:** Mother Infant Research Institute, Tufts Medical Center, Boston, MA 02184, USA

**Keywords:** placenta, lipid aldehydes, 4-HHE, 4-HNE, fatty acid

## Abstract

**Simple Summary:**

Omega-3 fatty acids are vital for optimal fetal and placental development. Under conditions of oxidative stress, toxic by-products called lipid aldehydes are produced, which can lead to inflammation. Little is known about the effects of lipid aldehydes on the placenta. We measured the effect of increasing the concentrations of two well-known lipid aldehydes, 4-hydroxynonenal and 4-hydroxyhexenal, on lipid metabolism genes in placental tissue. We found that they differentially impact fatty acid synthesis and uptake pathways in human placenta, which may have implications for the efficacy of omega-3 fatty acid supplementation in environments of oxidative stress.

**Abstract:**

Long chain polyunsaturated fatty acids (LCPUFAs), such as the omega-6 (n-6) arachidonic acid (AA) and n-3 docosahexanoic acid (DHA), have a vital role in normal fetal development and placental function. Optimal supply of these LCPUFAs to the fetus is critical for improving birth outcomes and preventing programming of metabolic diseases in later life. Although not explicitly required/recommended, many pregnant women take n-3 LCPUFA supplements. Oxidative stress can cause these LCPUFAs to undergo lipid peroxidation, creating toxic compounds called lipid aldehydes. These by-products can lead to an inflammatory state and negatively impact tissue function, though little is known about their effects on the placenta. Placental exposure to two major lipid aldehydes, 4-hydroxynonenal (4-HNE) and 4-hydroxyhexenal (4-HHE), caused by peroxidation of the AA and DHA, respectively, was examined in the context of lipid metabolism. We assessed the impact of exposure to 25 μM, 50 μM and 100 μM of 4-HNE or 4-HHE on 40 lipid metabolism genes in full-term human placenta. 4-HNE increased gene expression associated with lipogenesis and lipid uptake (ACC, FASN, ACAT1, FATP4), and 4-HHE decreased gene expression associated with lipogenesis and lipid uptake (SREBP1, SREBP2, LDLR, SCD1, MFSD2a). These results demonstrate that these lipid aldehydes differentially affect expression of placental FA metabolism genes in the human placenta and may have implications for the impact of LCPUFA supplementation in environments of oxidative stress.

## 1. Introduction

Maternal long chain polyunsaturated fatty acids (LCPUFAs) are essential for placental function and optimal fetal growth and development. During pregnancy, LCPUFAs are absorbed by the placenta, where they are metabolized, stored in the plasma membrane or lipid droplets and/or transferred to the fetus. LCPUFAs, particularly the omega-3 (n-3) fatty acid, docosahexanoic acid (DHA; 22:6, n-3), are essential for normal fetal neurological and cardiovascular development, and deficiencies in these LCPUFAs negatively affect long-term outcomes in behavior and body composition [1,2,3,4,5]. As fetal LCPUFA synthesis is inadequate to meet its substantial demand, the fetus depends on maternal LCPUFA intake and placental transfer. Thus, the dietary intake and placental metabolism of these fatty acids are highly important [6]. LCPUFAs, particularly the n-3 LCPUFAs such as DHA, have numerous beneficial effects owing to their anti-inflammatory and antioxidant properties; however, LCPUFAs are sensitive targets of oxidative stress, and their oxidation generates numerous toxic electrophilic compounds [7]. This can make their by-products less anti-inflammatory and create a more pro-inflammatory environment [8]. The impact of an adverse maternal condition, such as obesity, is associated with elevated levels of oxidative stress, which may alter the effectiveness and/or activity of these LCPUFAs. The growing number of women supplementing with n-3 LCPUFAs during pregnancy (38% in one US hospital, Ref. [9]) raises an important clinical question about the effect of LCPUFAs and their by-products on placental function.

The highly unsaturated nature of LCPUFAs makes them vulnerable to oxidation. Lipid aldehydes originate from the peroxidation of n-3 and n-6 LCPUFAs. Two major lipid aldehydes are nine-carbon 4-hydroxynonenal (4-HNE), a metabolite of arachidonic acid (20:4, n-6), and six-carbon 4-hydroxyhexenal (4-HHE), a metabolite of DHA. These oxidation by-products are highly reactive compounds and are involved in the etiology of many pathological processes, including obstetric conditions such as preeclampsia [10]. Under normal conditions, these aldehydes undergo detoxification via biotransformation and are eliminated from the body; however, due to their significant electrophilic properties, they can also react with a variety of biomolecules, including phospholipids, proteins and nucleotides, forming covalent adducts [7] and causing cell damage and programmed cell death [11]. 4-HNE and 4-HHE physiological concentrations are in the submicromolar range (<0.1–0.3 μM), while in conditions of extreme oxidative stress, 10 µM and up to 5 mM concentrations have been observed [12].

Within the first trimester of pregnancy, a burst of oxidative stress occurs during the establishment of maternal blood flow as part of normal placental development; however, it is proposed that a lack of antioxidant defense may lead to preeclampsia or even early pregnancy failure. In later pregnancy, studies show oxidative stress and lipid peroxidation are involved in the pathogenesis of multiple complications, including preeclampsia [13] and fetal growth restriction [14], and may be a factor in placental aging [15]. Although the placenta plays a key role in mediating the consequences of oxidative stress on the fetus, long-term programming can occur, not only in the feto-placental unit, but in alterations of physiologic systems such as heart and blood vessels, which can ultimately lead to cardiovascular disease in later life [16]. Therefore, it is highly important to understand the consequences of placental exposure to oxidative by-products such as 4-HHE and 4-HNE.

4-HNE and 4-HHE are elevated in many tissues in obese humans [17,18,19], as well as in porcine and rat placenta [20,21], and 4-HNE has been shown to activate peroxisome-proliferator-activated receptor (PPAR) pathways. Both PPAR-β/δ [22] in preadipocyte cells and PPAR-γ [23] in adipocyte cells are increased by 4-HNE, thereby upregulating lipogenic pathways. Less is known about the effects of 4-HHE on these pathways, and little to nothing is known about their effects within the placenta. Placental lipid metabolism is highly important for nutrient transfer to the fetus and overall placental function [24]. We and others have previously found that obesity, a condition characterized by oxidative stress, impacts these pathways in a lipotoxic manner (increased lipid esterification and storage, decreased mitochondrial function and β-oxidation) within the placenta [25,26]. These changes in lipid metabolism pathways in the placentas of patients with obesity coincide with significant alterations in the expression of PPARα and PPAR-γ [25]—master regulators of lipid metabolism [27,28,29,30,31]. We have further shown that supplementation with n-3 LCPUFA in the form of fish oil reduces placental lipid storage and lowers expression of lipogenic pathways such as PPAR-γ and its targets in the placentas of patients considered overweight and obese [32]. Considering the previously described effects of 4-HNE on PPARs [20,21] in other tissues, we hypothesized that lipid aldehydes may impact placental lipid metabolism pathways; however, the effect of these lipid aldehydes on placental lipid metabolism and their relationship with obesity in pregnancy is unknown. We sought to understand the effects of oxidative stress-related concentrations of 4-HHE and 4-HNE on lipid metabolism using full-term human placental explants.

## 2. Materials and Methods

### 2.1. Sample Collection

All samples and data were collected from patients delivering by scheduled caesarean section at Tufts Medical Center (Boston, MA, USA). Written and informed consent was obtained prior to study participation (IRB #13241). Maternal clinical data (including body mass index (BMI), maternal age, smoking status and race/ethnicity) were collected at time of consent. Maternal BMI ranged from 21.5 to 30.8 kg/m^2^ at time of first prenatal visit/end of first trimester. Maternal demographics data are listed in Appendix A Table A1. Exclusion criteria included known fetal anomalies, multiple gestations, hypertension, pre-existing diabetes, substance abuse, smoking or other comorbid diseases that could impact metabolism. Placental tissue from six pregnancies was collected within 2 h of delivery.

### 2.2. Placental Explant Isolation and Culture

Fresh tissue was collected into warm PBS from multiple areas across the maternal surface of the placenta following removal of the decidua. Placental villous explants were dissected as previously described [25]. All sample manipulations and incubation conditions were at 37 °C, and exposures were performed in triplicate. Placental explants were briefly acclimatized to culture medium supplemented with 10% fetal bovine serum, 1% Penicillin/Streptomycin and 0.2% ascorbic acid for 30 min, and then incubated for 24 h in the presence of a 0.15% ethanol (Thermo Fisher Scientific, Waltham, MA, USA) vehicle or 25, 50 or 100 µM of 4-HHE or 4-HNE (Cayman Chemicals, Ann Arbor, MI, USA), in addition to the control samples. After 24 h in culture, primary human placental explants begin to undergo modifications that may impact our assessments; therefore, longer incubations were not used [33]. Samples for gene expression analysis were snap frozen in liquid nitrogen and stored at −80 °C until assayed. Under physiological conditions, the cellular concentration of 4-HNE ranges from 0.1 to 0.3 μM; however, under conditions of oxidative stress, it accumulates up to concentrations of 10 μM–5 mM [12,34,35]. Lactate dehydrogenase (LDH) assay (Thermo Scientific, Waltham, MA, USA) was used to assess the cytotoxicity of ethanol and 4-HHE/4-HNE at 25, 50 and 100 µM.

### 2.3. Placental Gene Expression Analysis

Total RNA was obtained following homogenization of ∼50 mg of placental tissue in TRIzol reagent (Thermo Fisher), per the manufacturer’s guidelines, as previously described [36]. Triplicate explants were combined and extracted together. RNA quality was assessed for each sample by visualizing ribosomal RNA via gel electrophoresis and quantified and assessed for purity using a Nanodrop OneC (Thermo Scientific) spectrophotometer. Intact samples with 2 visible bands at 18S and 28S and whose 260/280 and 260/230 absorbance readings measured above 1.7 were included for gene expression analysis. A custom Nanostring nCounter Elements Panel (NanoString Technologies, WA, USA), as used previously [36], was used to assess the expression of 40 lipid metabolism genes (Appendix A Table A2) and an additional 3 housekeeping genes (L19, β-actin and Ywhaz). These genes were analyzed using the nCounter system, and assays were completed according to manufacturers’ instructions using 140 ng of RNA and a 24 h hybridization at 67 °C. Data were analyzed using the proprietary nSolver software and the ROSALIND platform (NanoString Technologies, Seattle, WA, USA) and normalized to internal controls and housekeeping genes to correct for major sources of error, including pipetting errors, instrument scan resolution, batch variations and sample input variability.

### 2.4. Statistics

Data were normalized using NSolver analysis software (Version 4.0.70), including nCounter Advanced Analysis (Version 2.0.115). Data were then analyzed by Friedman’s non-parametric 1-way ANOVA, followed by Dunn’s multiple comparisons test (GraphPad Prism version 9). A value of *p* < 0.05 was considered statistically significant. All data are presented as means ± SEMs unless noted otherwise.

## 3. Results

Maternal demographic data are presented in Appendix A Table A1. All mothers were non-smokers. There were no significant differences between gene expression in control explants (no vehicle or drug) and explants exposed to the drug vehicle. Based on the LDH activity, there was no placental cytotoxicity due to exposure to the vehicle or any concentration of 4-HHE or 4-HNE (data not shown).

### 3.1. 4-HHE Exposure Decreases Fatty Acid Uptake and Synthesis-Related Genes

To determine the effect of oxidized DHA on placental lipid pathways, we exposed placental explants to increasing concentrations of 4-HHE and analyzed the expressions of 37 lipid metabolism genes using the Nanostring Elements Platform. The expressions of five genes were significantly lower (*p* < 0.05) in placental explants treated with at least 50 µM of 4-HHE (Figure 1). The expression of genes associated with both fatty acid uptake (major facilitator superfamily domain-containing protein 2 (MFSD2a) and low-density lipoprotein receptor (LDL-R)) and fatty acid synthesis (sterol regulatory element-binding protein (SREBP)1, SREBP2 and stearoyl-coenzyme A desaturase (SCD1)) was decreased. The expression of all five was decreased with treatment using 100 µM 4-HHE: MFSD2a (*p* = 0.0052), LDL-R (*p* = 0.001), SREBP1 (*p* = 0.0024), SREBP2 (*p* = 0.0052) and SCD1 (*p* = 0.011). All gene expressions except SREBP2 were decreased with 50 µM 4-HHE: MFSD2a (*p* = 0.0024), LDL-R (P0.0417), SREBP1 (*p* = 0.0417) and SCD1 (*p* = 0.0417).

### 3.2. 4-HNE Exposure Increases Fatty Acid Synthesis and Transport-Related Genes

To determine the effects of oxidized AA on placental lipid pathways, placental explants were treated with increasing concentrations of 4-HNE, and the same lipid metabolism genes were analyzed using the Nanostring Elements Platform. The expressions of four genes were significantly increased (*p* < 0.05) in placental explants exposed to as little as 25 µM 4-HNE (Figure 2). The expressions of genes associated with both fatty acid synthesis (fatty acid synthase (FASN), acetyl CoA carboxylase (ACC)), fatty acid transport (fatty acid transport protein (FATP4)) and cholesterol esterification (acetyl CoA acetyltransferase (ACAT1)) were increased. The expressions of all were increased with treatment using 100 µM 4-HNE: FASN (*p* = 0.0052), ACAT1 (P0.011), ACC (*p* = 0.0417) and FATP4 (*p* = 0.0052). FASN expression was also increased with 25 µM 4-HNE treatment (*p* = 0.0219).

## 4. Discussion

The main goal of this study was to understand the effects of lipid peroxidation products on placental lipid metabolism gene pathways. While the effects of lipid peroxidation in other tissues, such as the heart [37], are understood to contribute to impaired function and underpin the pathophysiology of numerous diseases, the effects in placenta are not well studied. Our key finding was that 4-HHE and 4-HNE have differential effects on the expression of key lipid metabolism genes in the placenta. Interestingly, there was also no overlap of genes impacted by these two peroxidation by-products, indicating their unique impact on the placenta.

The number of patients supplementing with n-3 LCPUFA during pregnancy raises an important clinical question about the effect of LCPUFA and its by-products on placental lipid metabolism. We found that randomizing obese women to n-3 supplementation during pregnancy led to a 30% decrease in placental lipid content [32], consistent with an inhibition of esterification pathways. In the same cohort, supplementation with fish oil was also found to reduce placental inflammation [38]. It is unknown how the generation of oxidized by-products may impact these functions. Based on our findings, 4-HHE exposure decreased MFSD2A gene expression—the putative DHA transporter [39]—which may reduce transport of DHA into the cell. This is consistent with previous findings that high maternal DHA levels were associated with low MFSD2a in the placenta, and it is perhaps why we see decreased DHA in the cord blood of women with gestational diabetes [40,41]. We speculate that 4-HHE, as the oxidized product of DHA, may partially mediate this effect. It would be beneficial to investigate the effects of DHA vs. 4-HHE on MFSD2a expression in order to understand these mechanisms further.

4-HHE exposure inhibited the expression of lipid synthesis-related genes. Both SREBP genes, SREBP1 and SREBP2, are intimately involved in lipid synthesis. SREBP1 preferentially enhances the transcription of genes required for fatty acid synthesis and uptake, including ACC, FASN, SCD1 and lipoprotein lipase (LPL), while SREBP2 preferentially activates cholesterol synthesis. HHE reduced both SREBP1 and SREBP2 gene expression, which may lead to a reduction in lipogenesis. These findings are consistent with the impact of HHE’s unmodified parent fatty acid, DHA. DHA and its sister n-3 LCPUFA, eicosapentanoic acid (EPA), inhibit FA esterification via the lipogenic transcription factor SREBP1 [42], decreasing the expression of SCD1, an enzyme that modifies fatty acids destined for TG incorporation [43,44,45]. We previously reported that in trophoblasts, DHA and EPA inhibit fatty acid esterification, consistent with our in vivo findings that a high maternal n-3/n-6 LCPUFA ratio correlated with lower expression of placental PPARγ, diacylglycerol O-acyltransferase 1 (DGAT1, the final committed step in TG synthesis) and perilipin 2 (PLIN2, a marker of lipid droplet formation) [32]. Together, our new observations with 4-HHE, combined with previous data, suggest that DHA and its oxidized metabolite, 4-HHE (at physiological concentrations), have similar impacts: generally reducing placental lipogenic gene expression.

SREBPs also regulate the LDL receptor, which binds cholesterol-carrying LDL and promotes cellular uptake, including in the placenta, through receptor-mediated endocytosis. The decrease in SREBPs may explain the decrease in LDLR that we observed in our study following 4-HHE exposure. These changes may impact both the synthesis and uptake of cholesterol within the placenta. Impairments to cholesterol metabolism have previously been detected in oxidative stress environments such as the placentas of obese women [46]. Interestingly, exposure to 4-HNE (known to be increased with obesity [47]) increases expression of ACAT1, which is activated by cholesterol. ACAT1 converts free cholesterol to cholesterol esters for both storage and transport in plasma. If both products of LCPUFA peroxidation are produced during oxidative stress, 4-HHE and 4-HNE could both be limiting cholesterol uptake, along with storing and esterifying existing cholesterol. Cholesterol can be susceptible to peroxidation [48] as well as increasing lipid peroxidation [49,50], so reducing uptake and sequestering cholesterol could mitigate further placental damage.

Interestingly, 4-HNE exposure increased FATP4 gene expression, which is consistent with the overall function of 4-HNE in upregulating lipid uptake. This increase in FATP4 expression may be driven by an increase in the metabolism of long and very long chain FA and/or an elevated cellular requirement for FAs. A concomitant reduction in DHA uptake (if 4-HHE is also elevated) may exacerbate this need for FAs.

Acetyl-CoA carboxylases are enzymes that catalyze the carboxylation of acetyl-CoA to produce malonyl-CoA, and ACC expression was increased with 4-HNE exposure, which may indicate an increase in the first step of fatty acid synthesis. Malonyl-CoA is, in turn, utilized by FASN to produce long chain saturated fatty acids, such as palmitic acid, in a NADPH-dependent reaction [51]. Interestingly, FASN is also upregulated by 4-HNE, which would lead to increased palmitic acid biosynthesis. An excess of palmitate leads to inflammation and can cause lipotoxicity [52,53,54] via lipid accumulation, which can negatively impact placental function. As PPARγ is stimulated by oxidized lipids in term placentas and other tissues [55,56], the accumulation of 4-HNE may further increase esterification pathways, programming the placenta to preferentially store and accrue lipids. Palmitate overload can trigger lipid accumulation, insulin resistance, endoplasmic reticulum stress and oxidative imbalance, often concluding in cell death [54,57]. Though 4-HNE’s impact on the remainder of the pathway was limited, 4-HHE exposure decreased SCD1 expression, suggesting that if the two by-products are elevated in tandem, the new palmitic acid will be incorporated into other lipids such as phospholipids, triglycerides and cholesterol esters at a lower rate, depending on the comparative levels of the parent fatty acids.

The strengths of this study include its novelty. To our knowledge, this is the first study of the effects of these peroxidation products on placental lipid metabolism. This study suggests that 4-HHE and 4-HNE impact lipogenesis and FA uptake into the placenta, which may lead to detrimental outcomes such as inflammation. Future analysis of inflammation in these placental samples would be beneficial. Other strengths include our exclusion criteria and cytotoxicity assessment, which ensured a reliable baseline from which to study the effects of 4-HHE and 4-HNE exposure. The limitations of the study include the fact that we did not measure placental lipid content. Quantitative analysis by lipid type in these 4-HNE- and 4-HHE-exposed tissues would be beneficial to understand the effects of these gene changes on lipid synthesis, in particular cholesterol, palmitic acid and LCPUFAs. In addition, considering the changes in lipid metabolism that occur in the placenta from as early as 7 weeks from gestation [36], it would also be relevant to understand whether first-trimester placental tissue is sensitive to 4-HHE and 4-HNE. Future research should also address the effect of higher doses of 4-HNE and 4-HHE and their combined impact. We used doses in the physiological range and did not assess the effects of a more extreme environment. Although our results support the sensitivity of human placenta to 50 and 100 μM of both lipid aldehydes, we are likely missing the more dramatic effects seen in cases of extreme oxidative stress.

## 5. Conclusions

Our study demonstrates that lipid peroxidation by-products impact the gene expression of placental lipid metabolism pathways, possibly in a lipotoxic manner. It is important to note that the comparative effects of the two by-products of LCPUFA peroxidation would depend on the ratios of their respective parent fatty acids, arachidonic acid and docosahexanoic acid—n-6 and n-3 LCPUFAs (Figure 3). Overall, the impact on the genes involved in cholesterol and fatty acid synthesis and transport pathways may lead to a less efficient placenta, which may alter function and nutrient transfer to the fetus.

## Figures and Tables

**Figure 1 biology-12-00527-f001:**
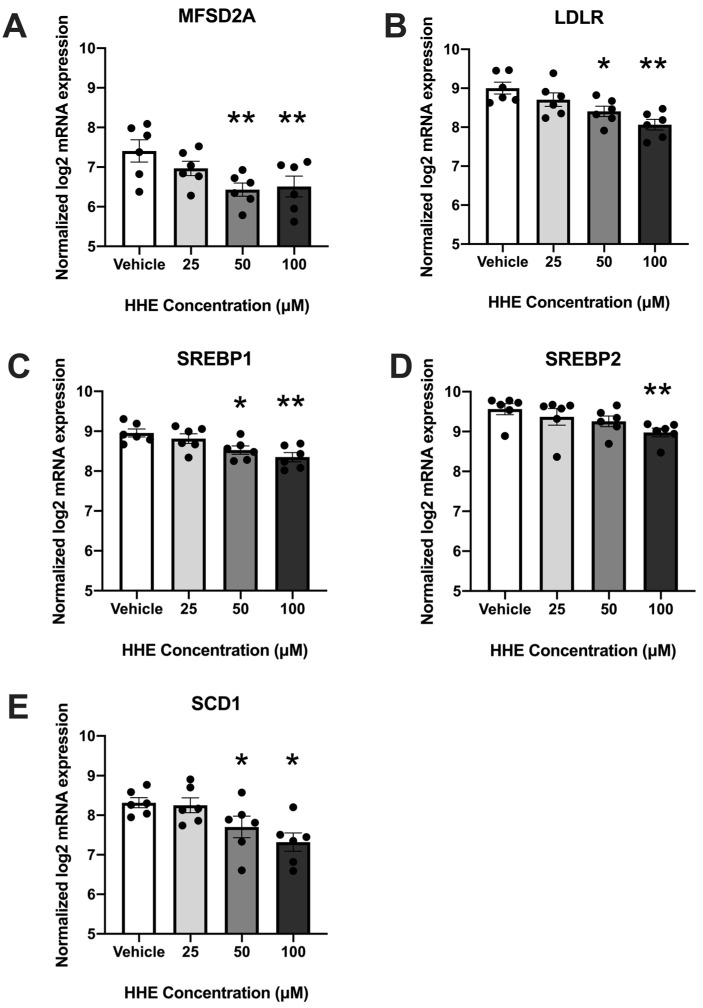
HHE exposure decreases fatty acid metabolism-associated genes in a concentration-dependent manner in human placental explants ((**A**), MFSD2a; (**B**), LDLR; (**C**), SREBP1; (**D**), SREBP2; (**E**), SCD1); *n* = 6. Data (means ± standard error of the mean) are expressed as normalized log2 mRNA expression. * *p* < 0.05, ** *p* < 0.01; data compared by non-parametric repeated measures, one-way ANOVA and Dunn’s multiple comparison test.

**Figure 2 biology-12-00527-f002:**
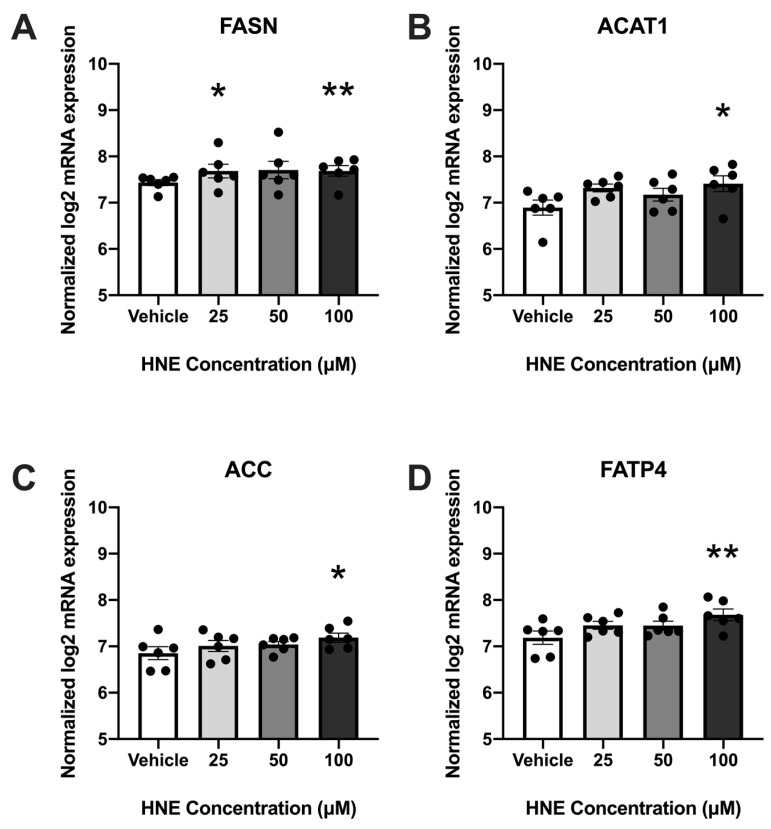
HNE exposure upregulates fatty acid metabolism-associated genes in a concentration-dependent manner in human placental explants ((**A**), FASN; (**B**), ACAT1; (**C**), ACC; (**D**), FATP4); *n* = 6/group. Data (means ± standard error of the mean) are expressed as normalized log2 mRNA expression. * *p* < 0.05, ** *p* < 0.01 vs. vehicle control by non-parametric repeated measures, one-way ANOVA and Dunn’s multiple comparison test.

**Figure 3 biology-12-00527-f003:**
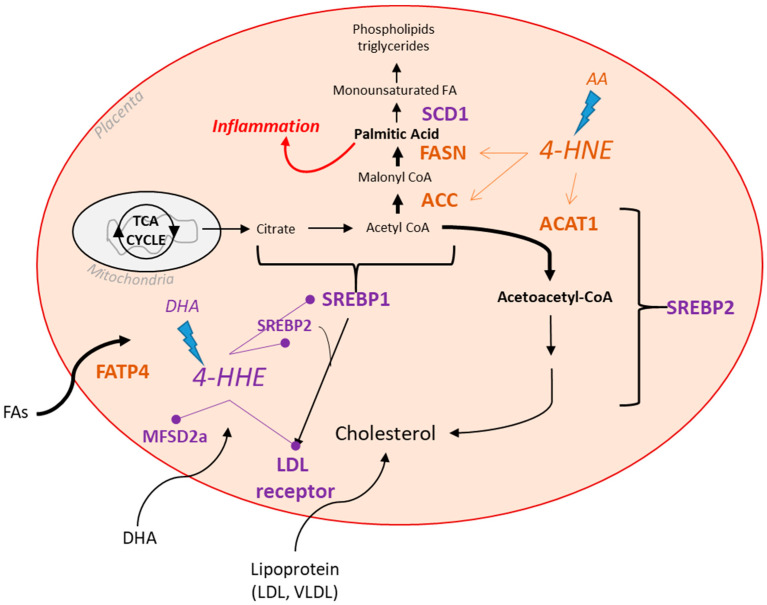
Summary and proposed model of the impact of 4-HHE and 4-HNE on placental lipid metabolism pathways. 4-HNE, an oxidative by-product of the n-6 LCPUFA, arachidonic acid (AA), drives an overall increase (orange) in lipogenesis and uptake genes. Increased ACC and FASN may produce more palmitic acid, whereas increased ACAT1 may heighten cholesterol synthesis. Higher levels of the fatty acid transporter FATP4 may increase the uptake of FAs. 4-HHE, an oxidative by-product of the n-3 LCPUFA, DHA, drives a decrease (purple) in the lipogenesis genes SREBP1 and SREBP2, which may decrease overall FA and cholesterol synthesis. Lower SREBPs may drive decreases in LDL receptors and cholesterol uptake. Low MFSD2a levels may reduce DHA uptake. Reduced SCD1 by 4-HHE impairs the last step in FA synthesis, potentially resulting in a build-up of the saturated palmitic acid, which may lead to the inflammation we see in placentas undergoing oxidative stress.

## Data Availability

Data are available upon request from the corresponding author.

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
