# Peer review of "Lipid Aldehydes 4-Hydroxynonenal and 4-Hydroxyhexenal Exposure Differentially Impact Lipogenic Pathways in Human Placenta"

_biology, 2023, doi:10.3390/biology12040527_

Round 1

Reviewer 1 Report

The authors present new data on gene expression changes in human placenta explants following exposure to a dose response of 4-HNE and 4-HHE. The role of oxidative stress on placental function and offspring outcomes is of critical importance and understanding the potential impact of lipid peroxidation in mediating some of the effects of oxidative stress in placental tissue is a logical next step. While the data presented in this article are straight forward and seemed to be completed with enough rigor, the significance/impact of these data are lacking because of this simplicity. In my comments below I suggest some additional experiments that would significantly strengthen the conclusions made in this paper and expand the overall impact of the authors findings.  If it is not feasible to complete such experiments then the authors will need to significantly soften their conclusions and point to the many limitations of interpreting such a limited analysis.  

Abstract:

Why did they choose to look at lipid metabolism genes. What was the rationale for examining that pathway?

“may have implications for efficacy of LCPUFA supp…” this statement does not seem to fit. Please remove or elaborate on the connection between your results and “efficacy of LCPUFA supp in environments of oxidative stress”.

Introduction:

Line 36: what long term outcomes? Please specify

Line 45: please elaborate on what you mean by “clinical consequences”. This statement is too broad and does not provide any meaningful information as it currently stands.

Line 52: please elaborate on what “pathological processes” again this is too broad to provide any meaningful information. If you move this sentence down one sentence to precede the “In pregnancy” it will likely provide the context I’m looking for.

Are there publications to support higher amounts of 4-HHE and/or 4-HNE in the placenta of obese patients? If so, this would be a helpful reference

Materials and Methods

Why did the authors choose a cohort of women with BMIs that spanned the overweight category. If you are making the argument that increased adiposity promotes oxidative stress, wouldn’t this variability in maternal environment induce too much variability across samples? How do you control for differences in the maternal environment? I appreciate the use of a primary culture system, but did the authors consider including any of the maternal factors in their analysis and/or using a cell culture model to confirm their findings and ensure that the effects are not somehow innate to the samples that were chosen?

Results

The dose response was mild with 4-HHE and non-existent with 4-HNE – making these data a bit unconvincing

The data may be muddied by the significantly different gene expression patterns that can be seen across all the cell types that exist within an explant sample. Confirming the gene expression changes in cell lines of different cell types and the subsequent protein changes with IHC to identify the cell types most effected by the lipid peroxidation products would significantly improve the conclusion that 4-HHE and 4-HNE are affecting these lipid metabolism pathways in a meaningful way.

Discussion

Line 175: The experiments in this study do not support or show what is stated in the first sentence in this paragraph. This needs to be revised/soften to represent what you truly did – examine gene expression. If you want to increase the relevance this work, then additional studies should take place to support the conclusion that 4-HHE inhibits lipid synthesis pathways – confirming changes in protein levels of SREBP targets for example would help. The fact that you didn’t see changes in lipogenesis transcripts speaks to the idea that 4-HHE does not actually impact lipogenesis… One would expect to see these downstream genes to also be impacted.

The discussion mentions the “potential” effects of having elevated 4-HNE and 4-HHE levels together in a physiological/pregnant setting. If it is likely to have both of these products at the same time, why didn’t the authors look at the effects of the combined treatment on gene expression? This approach would provide a more physiological understanding of the effects of lipid peroxidation in the placenta.

Line 229: Again the authors did not “confirm” that 4-HHE and 4-HNE do anything to lipogenesis and FA uptake in the placenta, they’ve only showed that these products can change gene expression. More experiments MUST be done in order to make these kinds of conclusions

Figure 3: This is a nice summary but is lacking the critical information of what cells in the placenta are affected by each of these pathways. Therefore, it is hard to interpret the big picture impact of what these authors have discovered, and may be misleading.

Minor Comments:

The authors start a lot of sentences with “This”. Please work to revise the manuscript to include more context in these sentences, it will help with the flow

The authors provide abbreviation for words in multiple places. Please revise to include the abbreviation at the first use and continue to use the abbreviation thereafter

Reviewer 3 Report

      The manuscript titled with ’Lipid aldehydes 4-hydroxynonenal and 4-Hydroxyhexenal exposure differentially impact lipogenic pathways in human placenta’ written by Aisha et al. was well written and clearly described. There are some questions about the study design etc. need to be answered to improve the manuscript.

1.     What was the dose selection basis for 4-HHE? And about 4-HNE, as Line 94-96 showed, under pathological conditions it ranged up till 5 mM, why the study design stopped at 100 μM?  

2.     Dose the placenta show different characteristic/capacity towards fatty acid uptake, synthesis and transportation along the whole pregnancy? Which stage is more important for the fetus development? Any proposals on how can we evaluate the other early stages or any relevant studies?

3.     Is 24 h exposure of the two investigated substances enough? What is the basis of the timepoint selection and what about the long-term effect?

4.     In Line 55-56, there were 3 references on the impact of impaired lipid synthesis during pregnancy on the baby, I think there should be more information on this, thus strength the meaning/basis of this study.

Round 2

Reviewer 1 Report

Thank you for addressing my concerns. While I do wish you were able to expand your experimentation to include some additional studies, I understand the challenges with this. I look forward to seeing future works expanding on more mechanistic studies.